# Survival of polycyclic aromatic hydrocarbon knockout fragments in the interstellar medium

Michael Gatchell [1,2 ✉], João Ameixa [2,3], MingChao Ji [1], Mark H. Stockett [1], Ansgar Simonsson [1], Stephan Denifl [2], Henrik Cederquist[1], Henning T. Schmidt [1] & Henning Zettergren [1]

Laboratory studies play a crucial role in understanding the chemical nature of the interstellar medium (ISM), but the disconnect between experimental timescales and the timescales of reactions in space can make a direct comparison between observations, laboratory, and model results difficult. Here we study the survival of reactive fragments of the polycyclic aromatic hydrocarbon (PAH) coronene, where individual C atoms have been knocked out of the molecules in hard collisions with He atoms at stellar wind and supernova shockwave velocities. Ionic fragments are stored in the DESIREE cryogenic ion-beam storage ring where we investigate their decay for up to one second. After 10 ms the initially hot stored ions have cooled enough so that spontaneous dissociation no longer takes place at a measurable rate; a majority of the fragments remain intact and will continue to do so indefinitely in isolation. Our findings show that defective PAHs formed in energetic collisions with heavy particles may survive at thermal equilibrium in the interstellar medium indefinitely, and could play an important role in the chemistry in there, due to their increased reactivity compared to intact or photo-fragmented PAHs.

[1] Department of Physics, Stockholm University, 106 91 Stockholm, Sweden. [2] Institut für Ionenphysik und Angewandte Physik, Universität Innsbruck, Technikerstr. 25, A-6020 Innsbruck, Austria. [3] Atomic and Molecular Collisions Laboratory, CEFITEC, Department of Physics, Universidade NOVA de Lisboa, 2829-516 Caparica, Portugal. ✉email: gatchell@fysik.su.se

There is little doubt that large polycyclic aromatic hydrocarbon (PAH) molecules are ubiquitous in the interstellar medium (ISM)[1–6]. PAHs were first proposed as carriers of unidentified infrared bands at 3.3, 6.2, 7.8, 8.6, and 11.3 μm originating from interstellar dust in the 1980s[7,8] and there are indications that they carry perhaps as much as 20% of the elemental carbon[3]. They are expected to form the basis for carbonaceous grains[9,10], act as catalysts for the formation of molecular hydrogen[11–16], and to be potential precursors for closed-caged fullerene molecules as large PAHs are broken down by energetic photons or particles[17,18]. However, the mechanisms leading to the formation of PAH molecules and fullerenes in interstellar and circumstellar environments are still not fully understood[19–22].

PAH molecules and ions are inherently stable systems with high ionization potentials and dissociation energies[23], which help explain their abundance in harsh interstellar regions[1]. However, the processing of these molecules by UV radiation from young stars[1] and by the impact of energetic particles from stellar winds and shockwaves[24,25] can partially break them down and increase their reactivity. This is one of the key steps in models aiming to describe the formation of fullerenes from large PAHs, and energy deposition is also required for certain $H_2$ production pathways involving PAHs[15,16]. When PAH molecules, or their fragments, are excited they can cool by dissociation, isomerization, or photon emission. Dissociation, a destructive process, will typically take place through the fragmentation channels with the lowest activation energies, i.e., through the loss of H, $H_2$, or $C_2H_2$ units, all of which require about 5 eV for intact PAH precursors[16,23]. Other fragmentation pathways, such as the loss of a single C atom (or $CH_x$ unit), require significantly more energy and are statistically unlikely in cases where the internal energy is spread over all internal degrees of freedom of the molecule, for instance in photo-driven processes[26]. However, there are mechanisms that may favor otherwise unexpected dissociation processes. One such example is collisions with atoms or ions[26,27]. An energetic ion colliding with a PAH molecule—or any other molecule for that matter—will not just transfer kinetic energy in the collision, but also momentum that is localized to the point of impact. This momentum transfer can in turn lead to individual atoms being knocked out of the molecule in a pool-ball-like scattering process[26–30].

Knockout-driven fragmentation has been identified in numerous experimental and theoretical studies of PAHs and fullerenes colliding with atoms and ions[27–29,31,32]. The most distinct fingerprint of these processes are fragments that have lost a single carbon atom, such as $C_{59}$ and $C_{23}H_x$ fragments from $C_{60}$ and coronene ($C_{24}H_{12}$) precursors, respectively[26,30]. In addition to these processes being observed with isolated molecules in the gas phase, knockout-driven fragmentation has also been shown to induce effective bond-forming reactions in cold, loosely bound clusters of such molecules that serve as laboratory analogs to small interstellar grains[33–35]. These grains may also be processed by energetic particles or radiation to form PAHs and fullerenes[36,37] as part of the carbon cycle of the ISM[2,38].

The quasielastic scattering processes that can lead to the knockout of atoms from a molecule dominate at impact velocities up to about 100 km/s[27], a common velocity range for stellar winds and shockwaves resulting from, e.g., supernovae[24,25]. It has therefore been proposed that knockout-driven reaction pathways may play an important role in the processing and evolution of complex molecules in the ISM (Fig. 1). But for these fragments to contribute to the chemistry of the ISM, they must survive long enough to either be processed further by energetic photons or particles or to react with another atom or molecule, which in astrophysical contexts often occur on timescales of years or more. The way in which they are formed is a violent process, but calculations show that PAHs or fullerenes missing a single C atom are thermodynamically stable on their own and thus will not fragment if they are in their lowest quantum states[26]. However, the internal energy of the fragments resulting from the initial collision may cause them to dissociate further before radiative cooling can stabilize the systems (typically the latter occurs on millisecond timescales[39]), destroying the product of the initial knockout process. Such secondary fragmentation processes often lead to less reactive products[26]. Previous experimental studies have demonstrated that fragments produced by carbon knockout are indeed stable on microsecond timescales[26,27], but until now it has not been possible to follow the fragments on radiative-cooling timescales (milliseconds or longer) in laboratory experiments.

Here, we report on the storage of coronene cations, and fragments of these ions produced in collisions with He atoms at velocities of 72 km/s, in the cryogenically cooled electrostatic storage ring DESIREE[40]. The ultra-low residual-gas density in the storage ring, coupled with the ability to dump the beam of circulating ions after any number of revolutions, enables us to follow small numbers of ions over long timescales. This allows us to study the cooling dynamics and stability of the PAH fragments for seconds or longer. We find that a small part of the stored ion population dissociates spontaneously on timescales up to 10 ms due to their high internal energy, but that the majority of the fragments formed by the knockout of a single C atom do not undergo additional spontaneous fragmentation and remain stable for the full 1-s measurement. At this point, the remaining ions have spontaneously cooled enough through photon emission to remain intact in isolation indefinitely. This shows that damaged PAH species formed in energetic collisions with atoms or ions can survive in the ISM and thus can contribute to astrochemical reactions there.

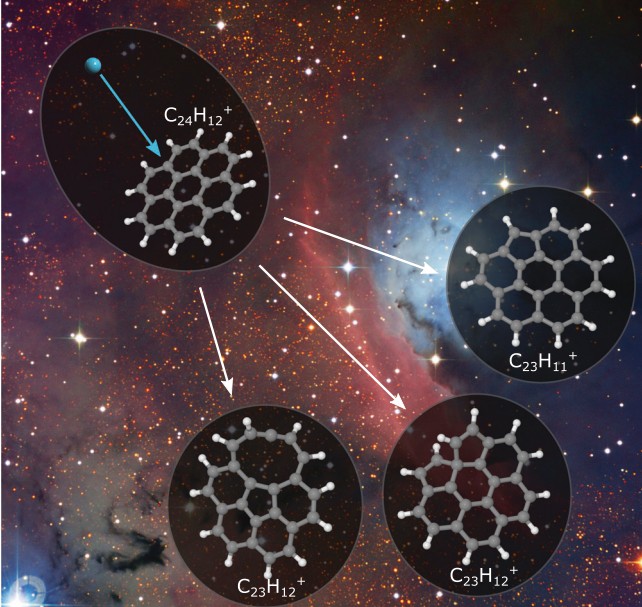

**Fig. 1 Examples of reaction pathways when a PAH molecule (here a coronene cation) is damaged by the knockout of a C atom in a collision with an energetic particle.** These types of reactions are proposed to take place in the ISM where PAH molecules/ions are abundant. The calculated $C_{23}H_x^+$ structures are from Stockett et al.[26]. Background image credit: ESO/U. G. Jørgensen.

## Results

**Fragment mass spectra.** A measured mass spectrum from colliding coronene cations with He gas at a center of mass energy of

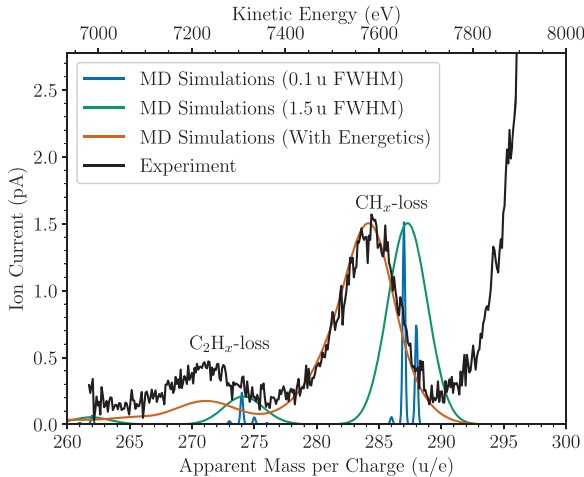

**Fig. 2 Mass spectra from collisions between coronene (cations, 300 u/e) and He at center-of-mass energies of 105 eV.** The experimental data is in black. The blue and green curves show the raw output from our molecular dynamics (MD) simulations, where one or more C atoms have been removed from the coronene molecule, with different resolutions (convolved with Gaussian profiles). The corresponding Full-Width Half Maximum (FWHM) values are given in the figure legend. The red data is the same as the green curve but includes information (from the simulations) on the translational motion of the products after the collisions. A detailed description of the simulations can be found in the Methods section. Source data are provided as a Source Data file.

105 eV is shown in Fig. 2. The resolution of this mass spectrum, which was obtained using the 90° electrostatic deflector system and ion optics on the injection line of DESIREE is $m/\Delta m \approx 200$, and thus not high enough to resolve products with different numbers of H atoms, but the peaks from fragments that have lost one or two C atoms, respectively, are clearly visible. The resolution is limited, in part, by the energy spread of the ions after the collisions. Since the measurements determine the kinetic energies of the ions, the mass scale is only a tentative scale to aid the reader. Also shown in the figure is a mass spectrum obtained from our classical MD simulations of the experimental conditions. A high-resolution mass spectrum (in blue, convolved using Gaussian profiles) from the simulations shows that the knockout of carbon atoms is the dominant fragmentation channel (after the loss of H atoms with no C-loss, not shown). The two most common fragments are those that have lost CH and C, respectively. There are additionally some fragments resulting from the loss of two or more C atoms, together with different numbers of H atoms. Reducing the resolution of the simulated spectrum to match the experimental one results in the green spectrum in Fig. 2, where the channels with different numbers of lost H atoms are no longer resolved. From the simulations, we also obtain the translational energies of the fragments parallel to the beam axis after the collisions. With this effect added to the simulated data (red curve, still with a 1.5 u FWHM resolution for each event) we see an excellent agreement with the position and shape of the $C_{23}H_x^+$ fragment peak with the experiments. The simulated data in Fig. 2 has been normalized so that the corrected C-loss peak matches the intensity from the experiments.

The experimental mass spectrum in Fig. 2 is in good agreement with previous studies of coronene colliding with He at center-of-mass energies close to 100 eV with the dominant fragment being the single C-loss product[26]. The next peak in the spectrum, that from the loss of two C atoms, is noticeably weaker in the theoretical spectrum than in the experiment. In the simulations, these fragments are solely the product of multiple knockouts by

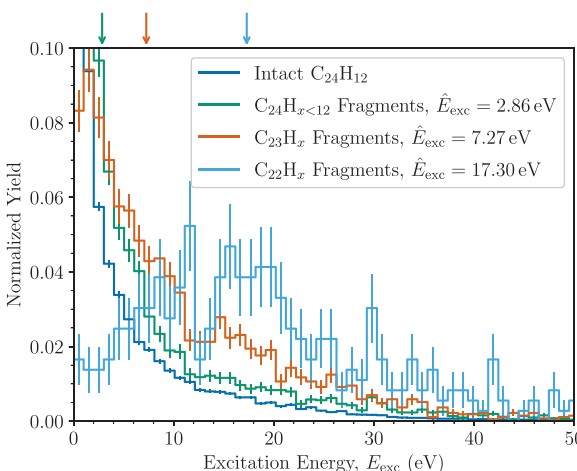

**Fig. 3 Excitation energy distributions for different fragments.** The values were obtained from our MD simulations of coronene molecules colliding with He atoms at a center-of-mass energy of 105 eV. The median of each distribution ($\hat{E}_{exc}$) is given in the legend and indicated by an arrow in the corresponding color at the top of the figure. The error bars represent the $1\sigma$ statistical uncertainties of each bin. Source data are provided as a Source Data file.

the single He projectile on sub-ps timescales. On the timescale of the measurement presented in Fig. 2—which is microseconds (μs), the time-of-flight between the collision cell and the analyzer—the loss of $C_2H_x$ due to delayed fragmentation of intact, internally hot coronene ions contributes to the peak in the mass spectrum in addition to secondary C-loss from $C_{23}H_x^+$ fragments. From the Electron Cyclotron Resonance (ECR) ion source, the ions are expected to have internal temperatures on the order of $10^3$ K, which is enough for some of the molecules to spontaneously fragment on μs timescales. In addition, the molecules may be heated in collisions with He. In Fig. 3, we show the extracted excitation energy distribution of different products from our MD simulations (not including the energy that the molecules carry from the ion source or the small electronic excitation energy from the collisions[41]). For all of the different fragments, the excitation-energy distributions are broad. The distribution for the intact ion includes distant interactions with little to no energy transfer, skewing it to lower energies depending on the maximum impact parameter used in the simulations (this is why no median energy loss is given in this particular case). Fragments that have lost only H atoms or a single C atom (with any number of H atoms) have distributions with similar shapes and median excitation energies of 2.9 and 7.3 eV, respectively (see the green and red lines in Fig. 3). The fragments that have lost two C atoms by prompt knockout on the other hand have much higher excitation energies, with a median of 17.3 eV and distribution that peaks between 10 and 20 eV. This is further indication that the $C_2H_x$-loss fragments in the experiment are predominantly not products of double-C knockout as the internal energies in the products would in most cases lead to further fragmentation on the μs timescale. Instead, we expect the majority of the $C_{22}H_x^+$ fragments injected into DESIREE (see below) to be the result of spontaneous $C_2H_2$ loss from intact coronene and/or secondary carbon loss from $C_{23}H_x^+$ fragments.

**Spontaneous decay of hot ions on millisecond timescales.** In Fig. 4, we show yields of neutral fragments resulting from the spontaneous decay of $C_{24}H_x^+$, $C_{23}H_x^+$, and $C_{22}H_x^+$ ions as a function of the storage time in DESIREE. The $C_{24}H_x^+$ ions were chosen by mass-selecting ions with masses of 300 u/e (8 keV

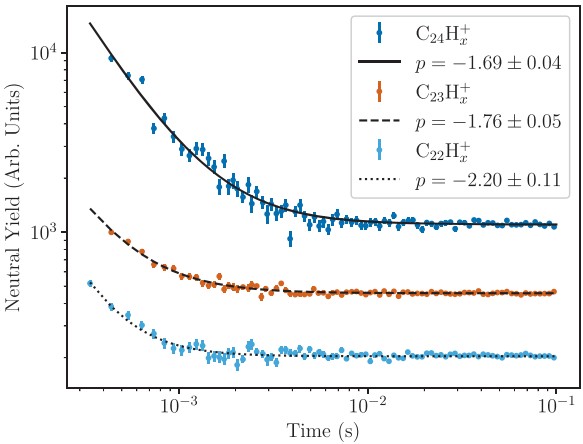

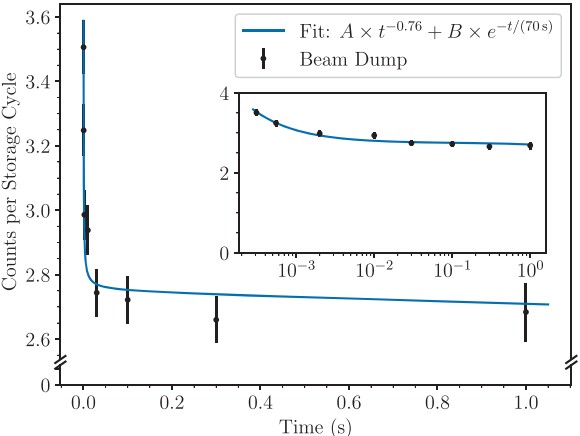

**Fig. 4 Neutral yields from the spontaneous decay of $C_{24}H_x^+$, $C_{23}H_x^+$, and $C_{22}H_x^+$ ions as a function of the storage time in DESIREE.** The curves show fits Eq. (1) and the respective exponents are given in the legend. The different data sets have been offset vertically to improve legibility. The constant levels reached after 10 ms are dominated by the detector background and are the same for all three measurements before they are shifted. Note the log-log scale. The error bars represent the 1$\sigma$ statistical uncertainties of each data point. Source data are provided as a Source Data file.

**Fig. 5 Averaged numbers of $C_{23}H_x^+$ fragments remaining in the storage ring determined by dumping the beams after different times.** This was achieved by switching the deflector at the end of a straight section to 0 V and counting the ions that then struck the detector. Each data point is the result of 500 storage and dump cycles. The solid blue curve shows the least square fit of Eq. (2). The same data with a logarithmic horizontal axis is shown in the inset. The error bars represent the 1$\sigma$ statistical uncertainties of each data point. Source data are provided as a Source Data file.

energy), while the two fragments were selected by tuning the 90° analyzer settings to the positions near the respective maxima measured in the mass spectrum in Fig. 2 (282 u/e and 271 u/e, respectively). All three data sets display a power-law decay as a function of time ($t$)

$$R(t) = A \times t^p + C, \qquad (1)$$

where $R$ is the measured rate of neutrals, and $A$ and $C$ are constants. The lines in Fig. 4 show fits to this functional form. Power-law decay curves over time are commonly observed from the spontaneous decay of ensembles of hot molecular ions or clusters with broad internal energy distributions and have been reported for a wide range of systems[39,42–48]. This typically occurs when a diverse ensemble of decaying units with many different time constants coexist. It has been shown that a $t^p$ behavior (with $p \approx -1$) can result from an ensemble with a broad internal energy distribution, which in turn gives a population with many different decay rates at a given time, $t$[42]. From the data and fits in Fig. 4, we see that the decay curves for $C_{24}H_x^+$ and $C_{23}H_x^+$ ions have nearly the same exponent ($p \approx -1.7$) within the statistical uncertainties. The $C_{22}H_x^+$ fragments on the other hand show a larger exponent with $p = -2.2$. For all three systems, this decay is only visible at short times, before the count rate reaches the level dominated by the constant detector background. From these measurements, we cannot draw any conclusions regarding the specific values of the exponents in Fig. 4 and they are merely presented as observations. However, the similarities between the decay curve for the $C_{24}H_x^+$ and $C_{23}H_x^+$ ions do suggest that the dissociation rate of the fragments is not significantly higher than for the precursor ions.

The only process that leads to neutral products as the molecular ions cools is fragmentation, which for PAH cations (and neutrals) most often takes place through the loss of H, $H_2$, or $C_2H_2$. But the measured rate can also be attenuated by radiative cooling, which is a non-destructive process that cannot be observed directly from neutral particle detection. Such a process can, however, lead to a deviation from a single power-law decay behavior by reducing the fragmentation rate at later times or, if the cooling rate by photon emission is higher than the fragmentation rate at all measured times, dictate the overall

shape of the decay curve. The disappearance of the decay signal after 10 ms does not imply that the storage ring is empty of ions, but is instead due to the remaining population of ions being too cold to spontaneously decay (at least at a detectable rate). In this latter part of the decay curve the signal—i.e., the measured rate of neutrals—is too low to be distinguished from the detector background level. We can therefore not conclude whether any ions are left in the ring at long times from the decay curves shown in Fig. 4 alone.

**Probing fragments at long times**. In order to investigate if $C_{23}H_x^+$ fragments produced by single C knockout are stable on longer timescales we periodically switched off one of the deflectors in the storage ring to dump the stored ions onto a detector after different times of storage. The results of these measurements are shown in Fig. 5 and cover nearly four orders of magnitude in time. The shortest storage time obtainable with this method is when the beam is dumped after only half of a turn in DESIREE ($t = 309$ μs after ion production) and the longest measured time involved dumping the ion beam one second after production. During this time we can see that the number of $C_{23}H_x^+$ ions present in the storage ring decreases from $3.51 \pm 0.08$ ions per storage cycle at the first measurement point to $2.68 \pm 0.09$ at the last. At short times the population of ions is depleted by the spontaneous fragmentation of hot ions, which gives the decay curves in Fig. 4. Strikingly, after approximately $10^{-2}$ s the decay abruptly stops as the population has cooled sufficiently such that the storage lifetime is limited by the experimental conditions, i.e., by collisions with the extremely dilute residual gas. Since the latter takes place at a rate proportional to the number of ions in the ring, it will result in an exponential decrease of the ion population with storage time. To identify the contributions from the two processes—dissociation of (some of the) stored ions and neutralization through collisions with the residual gas—we fit the following functional form to the data in Fig. 5:

$$N(t) = A \times t^q + B \times e^{-t/\tau}. \qquad (2)$$

Here, $N(t)$ is the number of ions remaining stored at time $t$, $\tau$ is the ion-beam storage lifetime, $q$ is the exponent of the power-law decay at short times, and $A$ and $B$ are constants (the value of $A$

here is related to, but not equal to, the value in Eq. (1)). In contrast to the earlier measurements of neutrals from the spontaneous decay shown in Fig. 4, which only shows the contribution of dissociating fragment ions, the measurements in Fig. 5 count all of the stored $C_{23}H_x^+$ ions at any given time. The measured rate of neutrals, $R$ in Eq. (1), can be expressed in terms of the total population $N$ by $R(t) \propto -dN/dt$. If we assume that the influence of residual gas collisions is negligible at short storage times, then the second half of Eq. (2) is constant at early times which gives $R(t) \propto t^{(q-1)}$, and that $p = q - 1$ (for $p < -1$) at early times. From the fit in Fig. 4, we found that $p = -1.76 \pm 0.05$ for the $C_{23}H_x^+$ ions. Since $p$ can be accurately determined from a large number of data points we choose to use this value to set $q = -0.76$ when fitting Eq. (2) to the measurements shown in Fig. 5. The storage lifetime of $C_{23}H_x^+$ was determined to be $\tau = (70 \pm 5)$ s from complementary measurements where the stored ions were dumped onto the detector after 100 s. The storage lifetime of ions in the DESIREE, which is typically on the order of minutes or longer[49], depends on the ring parameters and vacuum conditions and does not reflect the inherent stability of the ions. This value was used as a constant when fitting the data in Fig. 5 to Eq. (2).

The key finding from these measurements is that a significant fraction of the $C_{23}H_x^+$ ions injected into the storage ring remains stable for the storage lifetime of the experiment. From the fit we obtain $B = 2.77 \pm 0.03$, which corresponds to the mean number of detected ions in the ring (per injection) that are too cold to spontaneously decay. Of the ions that are initially injected into the ring (counted after half of a revolution to be $3.51 \pm 0.08$ on average), close to 80% remain after the first 100 ms. We can thus conclude that the population of stored ions that give rise to the neutral products at early times, measured in Fig. 4, only consists of about 20% of the total stored population. The rest of the ions would then circulate indefinitely if not for the collisions with residual gas particles in the experiments. In addition to the ions that make it from the collision cell to the storage ring, there may also be a population of short-lived, highly excited ions that fragment somewhere along the beamline between the collision cell and the last deflector before the detector.

We use our MD simulations to estimate the fraction of $C_{23}H_x^+$ fragments initially produced in the gas cell that would not dissociate at a later time in a collision-free environment. By assuming that all $C_{23}H_x^+$ ions with internal energies below the 7 eV dissociation energy of the most stable $C_{23}H_x^+$ isomer[26] do not dissociate, we see from the MD-simulations in Fig. 3 (red line) that 50% of the $C_{23}H_x^+$ fragments produced have energies below this limit. Previous studies of anthracene ($C_{14}H_{10}$) ions determined that approximately 10 eV of internal energy is required to induce fragmentation before the ions are stabilized by radiatively cooling[50,51]. If we use this higher value as an upper limit on the excitation energy $C_{23}H_x^+$ ions that survive on long timescales can carry, then we find that 60% of the population in Fig. 3 are below this limit. From this, we estimate that about half of the $C_{23}H_x^+$ ions formed in collisions with He in the gas cell (cf. Fig. 6) remain intact long enough to be stabilized by radiative cooling. In terms of cross-sections, this value gives an effective cross-section of $2 \times 10^{-16}$ cm$^2$ for producing surviving $C_{23}H_x^+$ products under the conditions studied here (collisions with He at ~70 km/s).

## Discussion

Here, we have shown that highly reactive fragments of PAH molecules damaged by the knockout of carbon atoms in collisions with energetic particles can remain stable on indefinitely long, i.e., astronomical, timescales in the gas phase. Extensive modeling work on the processing of PAH molecules by interstellar shocks

has been performed by Micelotta, Jones, and Tielens[24]. In that work the authors model the processing of PAH-molecules in supernova shockwaves over a range of molecular sizes and shock velocities. They found that the dominating PAH-damaging mechanism at shock velocities below 75 km/s is impact by He or H[24]. These are the conditions that are reproduced in our experiments and for which we show that the resulting PAH fragments will stabilize on millisecond timescales. Depending on the gas column density and PAH size used in their model, Micelotta et al. found that each individual PAH molecule will typically collide with one to two He atoms with enough energy to remove a carbon atom before they are injected into the ISM[24]. At higher velocities (above 100 km/s) the authors identified that additional collisions with electrons will contribute to damaging the PAH molecules further. However, large PAH molecules (containing a few hundred C atoms) will be less sensitive to this mechanism, so even there the knockout mechanism is expected to be important.

The precursor system that we have studied here is the coronene cation, $C_{24}H_{12}^+$. This well-studied molecule is commonly used as a prototype for laboratory astrophysics studies but is smaller than what is considered to be the typical PAH size in the ISM (50–100 C atoms)[2]. With increasing PAH size the fraction of molecules that survive having a C atom removed from their structures will only increase. This is because the energy required to remove the atom will be independent of PAH size, as will the excitation energy remaining in the fragment, but the number of internal degrees of freedom in the molecule increases with size, reducing the rate of secondary fragmentation[26]. We can thus expect our findings to be directly applicable also for larger PAH species and related species such as fullerenes.

The knockout process produces molecular fragments that are unlikely to be formed, e.g., in photo-fragmentation where the loss of a single C atom from a PAH molecule essentially is energetically disfavored[26]. For PAH molecules these defects introduce new structural features, e.g., carbon rings consisting of fewer or more than six atoms as shown in Fig. 1, and an increased reactivity compared to the intact species[34,41]. In interstellar and circumstellar environments, the most probable species to react with PAH molecules damaged by carbon knockout would likely be hydrogen, carbon atoms or chains, or other hydrocarbon molecules (such as other PAH molecules and fragments)[1]. Molecular electronic structure calculations of intact and damaged PAH molecules have shown that the binding energy of an H atom to a large PAH molecule/cation (circumcoronene, $C_{54}H_{18}$) increases by a factor of four if a C atom is removed from the PAH molecule[41]. For other species, such as the aromatic phenyl radical ($C_6H_5$), the increase in binding energy for the damaged PAH is even greater[41]. The increased chemical reactivity of PAH molecules damaged by knockout processes could therefore play an important role in the growth of larger carbonaceous molecules and dust particles.

Another aspect is related to the formation of fullerene molecules in astronomical environments. One of the important steps in modeling fullerene formation from large PAH precursors is the conversion of hexagonal rings in the planar molecules to the pentagonal rings that are necessary to form a closed cage structure. This has been demonstrated to take place when photoprocessing large PAH species[17,52], but clearly, the knockout of individual C atoms by energetic particles, which is a highly efficient process, could very effectively lead to similar features. It is not yet clear what the relative importance of these different mechanisms are though, but further modeling or the identification of characteristic spectral features will be required to determine their potential contributions. The same goes for the increased chemical reactivity of damaged PAHs where the

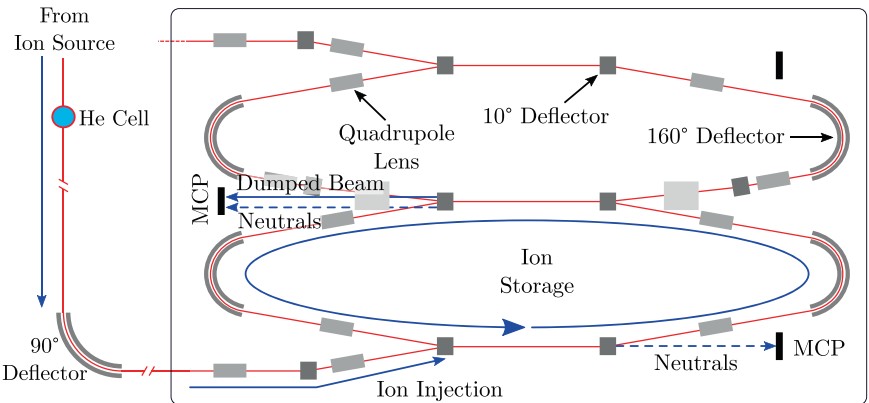

**Fig. 6 Overview of the experimental setup.** Mass-selected coronene cations with kinetic energies of 8 keV are passed through a gas cell containing a dilute He gas in order to induce fragmentation in a small fraction of the $C_{24}H_{12}^+$ ions that pass the cell. Selected fragments ($C_{23}H_x^+$ or $C_{22}H_x^+$) or intact ions ($C_{24}H_{12}^+$) are stored (in separate measurements) in one of the DESIREE rings for stability measurements. The electrostatic beamline elements are indicated by gray boxes that are labeled in the upper ring. Only the lower ring is used in these experiments. Neutral products formed along the two straight sections are detected by microchannel plate (MCP) detectors. The cryogenic region is indicated by the thin black line enclosing both rings.

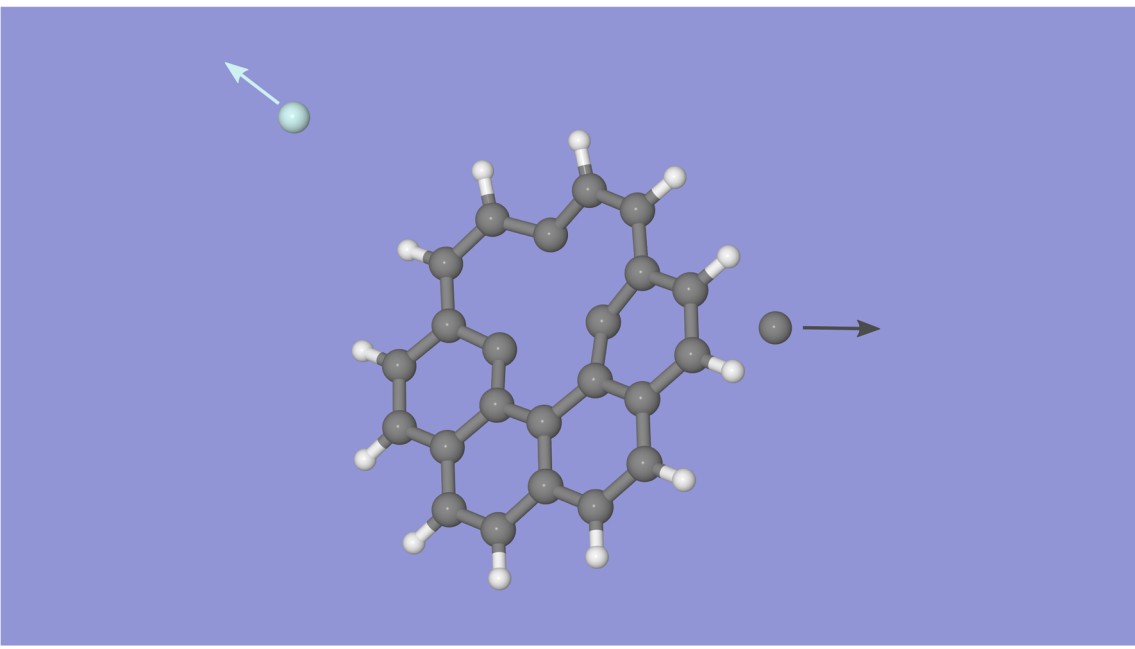

**Fig. 7 Snapshot of a simulated collision between He and coronene.** This frame shows the positions 35 fs after the start of the simulation (about 25 fs after the collision). The He projectile was fired from the left and has removed a single C atom from the PAH molecule and is now back-scattered. About 6 eV of internal energy is transferred to the $C_{23}H_{12}$ fragment in this particular collision. Supplementary Movie 1 shows the first 100 fs of the simulation in steps of 0.5 fs. A total of $10^5$ simulations with random orientations of the molecule and with randomized projectile trajectories were performed in the present study.

importance of such systems to chemical processes in the ISM remains to be seen. Nevertheless, it is clear that molecular fragments created by the knockout of atoms from complex molecules such as PAHs could indeed survive indefinitely in the gas phase, for example in the ISM.

## Methods

**Experiments**. The experiments were performed using the Double ElectroStatic Ion Ring ExpEriment (DESIREE) at Stockholm University[40,53]. DESIREE is outlined in Fig. 6 and consists of two electrostatic ion-beam storage rings, each with a racetrack layout and a circumference of 8.68 m, enclosed in a single cryogenically cooled vacuum chamber. The two rings share a common straight section to allow for merged-beams experiments with interactions of oppositely charged keV ion beams at close to zero relative velocity. The rings are operated at 13 K and with a residual gas density on the order of $10^4$ cm$^{-3}$ (mostly consisting of $H_2$), corresponding to a

pressure of about $10^{-14}$ mbar[40]. In the present experiments, only a single storage ring was used, as shown in Fig. 6.

Coronene powder (99.9% purity from Sigma Aldrich) was heated in an oven operating at about 230 °C, and the coronene vapor was fed into the plasma of an Electron Cyclotron Resonance (ECR) ion source. Ions extracted from the source were accelerated to 8 keV and chopped into pulses matching the length of the storage ring. Coronene cations were mass-selected using a 102° bending magnet (not shown in Fig. 6). A 7 cm long gas cell containing He with a pressure of $1.1 \times 10^{-2}$ mbar (chosen by optimizing for the knockout yield) was mounted on the beamline in order to produce coronene fragment ions from collision-induced dissociation. The 8 keV lab energy of the PAHs together with the He target resulted in collisions with energies in the $C_{24}H_{12}^+$ + He center-of-mass frame of 105 eV. This corresponds to a velocity of about 72 km/s, which is typical for gas in interstellar shocks[24] and stellar winds[54]. All fragments exiting the gas cell had close to the same velocity, so their kinetic energies scaled linearly with their masses. Products from the collision cell (intact ions or fragments) were mass/energy selected using a 90° electrostatic deflector system (consisting of 10°, 70°, and 10° deflectors in sequence) just prior to the injection of the ions into the storage ring. This deflector system was also used to measure a fragment

mass spectrum by scanning the deflection voltage together with the ion optics voltages along the beamline and measuring the ion current with a Faraday cup. When storing the ions, the settings on the deflectors and ion lenses in the storage ring were also adjusted accordingly from the setting optimized for the precursor beam depending on the mass-selected fragment. The first 10° deflector encountered by the ions that enter the ring was switched between injection and storage modes before an ion pulse made its first full revolution in the ring.

The stored ions circulated in the storage ring with a revolution frequency of 8.3 kHz. Neutral products, produced either by the spontaneous decay of internally hot ions or by collisions with residual gas, were not influenced by the electric fields and left the ring along a straight trajectory. Neutral products formed along either of the two straight sections were detected by microchannel plate (MCP) detectors located 1.2 m after the respective straight sections and independently counted as a function of the storage time. To probe the long time-stabilities of knockout fragment ions, the beam was dumped onto one of the detectors (see Fig. 6) after a predetermined time of storage to count the number of ions remaining in the ring. This was done by rapidly switching off the deflector at the end of the middle straight section and letting the stored ions continue straight onto the detector.

**Molecular dynamics simulations.** We have performed classical MD simulations of coronene molecules colliding with He atoms in order to determine cross sections for different knockout channels and for estimating the internal energy distributions of the ions that we are storing in DESIREE. The interactions between C and H atoms were described using the Tersoff potential, a reactive many-body potential that allows bonds to be formed and broken between these types of atoms in a realistic way, taking into consideration the bond order of each atom[55,56]. The atomic parameters and the mixing terms used with this potential are from Stockett et al.[57] Interactions between He and the atoms in coronene were described using the Ziegler–Biersack–Littmark (ZBL) potential[58]. The ZBL potential is a shielded Coulomb potential for describing the scattering of colliding atoms. The combination of these potentials has been successfully used in the past to simulate collisions between atoms and carbonaceous molecules like PAHs and fullerenes[27,32,34,59,60]. All of the simulations were performed using the LAMMPS software package[61].

The simulations were performed by first defining the structure of a coronene molecule, which is optimized to the potential-energy minimum in the Tersoff force field. In each simulation, the planar molecule is initially located in the $xy$-plane with its center-of-mass at the origin of a three-dimensional cartesian coordinate system. It is then randomly rotated around the origin at the beginning of each simulation to recreate the randomly oriented collisions in the experiment. A He atom projectile is initialized at $z = 7$ Å with the $x$- and $y$-coordinates individually randomized within a $12$ Å $\times 12$ Å box centered at the origin of the $xy$-plane. The simulation is started by firing the He atom in the negative $z$-direction with the same velocity as in the experiments (72 km/s), giving center-of-mass collision energy of 105 eV. Each simulation was followed for $5 \times 10^{-13}$ s, long enough to follow prompt knockout processes, using a time step of $5 \times 10^{-18}$ s. At the end of each simulation the positions and velocities of all atoms were recorded and analyzed to determine which bonds had been broken or formed, and the energetics of each fragment (both internal and kinetic energies of the fragments). A total of $10^5$ simulations were performed, each with different initial parameters. A snapshot from a single simulation can be seen in Fig. 7 and the complete first 100 fs can be seen in Supplementary Movie 1.

Using classical potentials, the simulations do not contain any description of the electron dynamics of the target or projectile. Likewise, the simulations do not contain any description of inelastic scattering resulting from interactions between the projectile and the electrons in the target molecule. At the velocities studied here, which are slow compared to the motion of bound electrons, this mechanism is relatively weak and plays only a small role in the overall energy transfer in the collisions[27]. The potentials are defined for neutral systems, but for large PAH molecules with many delocalized electrons the differences in binding energies between neutral molecules and cations are small, so this effect does not significantly affect the outcome of the simulations when compared to the experiments.

## Data availability
The raw data related to this paper is available from the authors upon reasonable request. Source data are provided with this paper.

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

## Acknowledgements

This work was performed at the Swedish National Infrastructure, DESIREE (Swedish Research Council Contract No. 2017-00621). M.G., M.S., H.C., H.S., and H.Z. acknowledge personal support from the Swedish Research Council (contracts 2020-03104, 2016-03675, 2019-04379, 2018-04092, 2020-03437, respectively). It is a part of the project "Probing charge- and mass-transfer reactions on the atomic level", supported by the Knut and Alice Wallenberg Foundation (Grant no. 2018.0028). J.A. acknowledges the Portuguese National Funding Agency FCT-MCTES through scholarship grant number PD/BD/114447/2016, as well as the Radiation Biology and Biophysics Doctoral Training Program (RaBBiT, PD/00193/2012); UID/Multi/04378/2019 (UCIBIO); UID/FIS/00068/2019 (CEFITEC). This article is based upon work from COST Action CA18212—Molecular Dynamics in the GAS phase (MD-GAS), supported by COST (European Cooperation in Science and Technology). We thank the storage ring operators and technical staff at the Department of Physics of Stockholm University for their support in these experiments.

## Author contributions

H.Z., H.C., H.S., M.G., and M.S. conceived this work. A.S. developed the new experimental methods used in this work to store the fragment ions. M.G., J.A., MC.J., M.S., A.S., H.Z., and H.S. prepared and performed the experimental measurements. M.G. analyzed the data and performed the theoretical simulations. H.C., H.S., H.Z., and S.D. provided resources for the study. M.G., H.Z., H.C., and H.S. prepared the paper, which all authors reviewed, discussed, and approved.

## Funding

## Competing interests

The authors declare no competing interests.
