## [Peer Review File · Nature Communications]

REVIEWERS' COMMENTS

Reviewer #1 (Remarks to the Author):

The manuscript by Gatchell et al. describes an ion-beam storage ring experiment to explore the fragmentation of coronene bombarded with He atoms and places the results in an astrophysical context. I find the work to be of exceptionally high quality, and commend the authors for a clear, well presented manuscript. The work is timely, the interpretation of the results is well-supported by the experimental data, and the results themselves seem to be of broad interest and applicability. In my opinion the discussion of the astrophysical implications is sufficient to provide context and suggest reasonable areas for follow-up by other experts while not straying into idle speculation.

This is the first manuscript in a decade I am pleased to offer a recommendation of publishing as-is without further revision. This manuscript was a joy to read.

Reviewer #2 (Remarks to the Author):

This is an interesting paper that exploits the characteristics of the DESIREE storage ring to ascertain if fragments of PAHs induced by dissociative impact by helium atoms (present in shock fronts) can be retained in the ISM for long periods and hence play a role in further chemistry in those regions. The experiments are well controlled and characterized such that their validity is without question. The authors enhanced report on the relevance to Astrochemistry is welcome addition as is the report on MD simulations. I recommend publication.

Reviewer #3 (Remarks to the Author):

In this manuscript, Gatchell and coworkers combine a classical collision experiment with a state-of-the-art storage ring study. In a series of previous publications, the Stockholm group had already proven the important role of knock-out processes in such collisions and thoroughly studied the resulting fragmentation processes. The relevance of knockout processes for the field of astrochemistry is obvious: PAHs or PAH ions which lack a single carbon atom at a central site are very reactive species. As the authors correctly state, these molecules could play a key role for the growth of PAHs, dust particles and even fullerenes in the interstellar medium.

The problem with the astrochemical application has been, that we simply didn't know if the reactive species formed in knock-out collisions would be stable on astronomical timescales. By selectively injecting knocked-out coronene cations into the electrostatic and cryogenic DESIREE storage ring, the authors were able to thoroughly study their lifetime over very long timescales. The exciting result is presented in this manuscript: A small fraction of molecules decays via fragmentation processes on a few milliseconds timescale. However, about 80% of the knocked-out PAHs are not subject to this decay channel. Instead, they are only decaying via collisions with residual gas molecules. The latter decay process is unrelated to the inherent stability of a knocked-out PAH. The authors therefore conclude that the majority of PAH knock-out collisions in the ISM will most likely result in very reactive species that are stable on astronomical timescales.

I consider this manuscript very exciting for two reasons: On the one hand, the selective investigation of ion-atom collision product in a cryogenic electrostatic storage ring is an elegant and technically challenging endeavor. On the other hand, the astrochemical implications are obvious: Knocked-out PAHs do form in the ISM, they are stable on astronomical timescales and as a consequence they likely play an important role in molecular growth processes in the ISM.

The manuscript is well written and accessible for a broad audience. I recommend publication in its current form.